# Implementation of Deep Deterministic Policy Gradients for Controlling Dynamic Bipedal Walking

**DOI:** 10.3390/biomimetics4010028

**Published:** 2019-03-22

**Authors:** Chujun Liu, Andrew G. Lonsberry, Mark J. Nandor, Musa L. Audu, Alexander J. Lonsberry, Roger D. Quinn

**Affiliations:** 1Department of Mechanical and Aerospace Engineering, Case Western Reserve University, Cleveland, OH 44106, USA; cxl936@case.edu (C.L.); agl10@case.edu (A.G.L.); mjn18@case.edu (M.J.N.); ajl17@case.edu (A.J.L.); 2Department of Biomedical Engineering, Case Western Reserve University, Cleveland, OH 44106, USA; mxa93@case.edu

**Keywords:** biped, DDPG neural network, gait, stability

## Abstract

A control system for bipedal walking in the sagittal plane was developed in simulation. The biped model was built based on anthropometric data for a 1.8 m tall male of average build. At the core of the controller is a deep deterministic policy gradient (DDPG) neural network that was trained in GAZEBO, a physics simulator, to predict the ideal foot placement to maintain stable walking despite external disturbances. The complexity of the DDPG network was decreased through carefully selected state variables and a distributed control system. Additional controllers for the hip joints during their stance phases and the ankle joint during toe-off phase help to stabilize the biped during walking. The simulated biped can walk at a steady pace of approximately 1 m/s, and during locomotion it can maintain stability with a 30 kg·m/s impulse applied forward on the torso or a 40 kg·m/s impulse applied rearward. It also maintains stable walking with a 10 kg backpack or a 25 kg front pack. The controller was trained on a 1.8 m tall model, but also stabilizes models 1.4–2.3 m tall with no changes.

## 1. Introduction

Spinal cord injuries (SCI) can cause paralysis, resulting in minimal motor control and rendering standing and walking impossible. Exoskeletons can help patients regain their ability to stand and walk on their own. It has been established previously that combining functional neuromuscular stimulation (FNS) with a powered, lower limb exoskeleton can restore locomotion to such individuals [1,2,3].

There remain many challenges in realizing such systems, given that each patient’s body is unique. One of the primary problems needing more work is the generation of adaptive control systems for stable walking and fall prevention. While much research has been invested in such control for legged robots, there have been few applications of these methods to exoskeletons.

The design of algorithms for control of bipedal robot locomotion is a topic of intense research interest [4,5,6,7,8] and several different methods have been developed. Some of these focus on the concept of finding a zero-moment point (ZMP) about which to step. Kim and Oh [9] reported on a controller using a ZMP-based technique with feedback from inertial sensor measurements. This controller has three subcomponents: one that adjusts a pre-defined walking pattern, a second one for balance in real-time using information from sensory feedback, and a third one for motion control based on previous experience. The controller is successfully implemented on a robot that walks without the need for support and without any extensive tuning of the controller parameters. A similar method [10], again featuring a controller comprised of three modules, demonstrates bipedal stability using a ZMP method. The three modules perform body inclination control, ZMP control, and foot adjustment control. The last component is primarily invoked when the system encounters uneven terrain. The ZMP reported by Yokoi et al. [10] is computed using torque sensors at the ankles and controlled via an adjustment of the orientation of the trunk. In implementation, the robot can walk stably with a step length of 0.2 m/step and a step period of 0.8 s/step. A key difference between these two works is in how the foot position is chosen. In Kim and Oh [9], foot placement is defined as part of the desired motion pattern, while, in Yokoi et al. [10], it is computed by inverse kinematics from the desired joint angle trajectories.

In this paper, based on the concept of the ZMP, a simplified, but robust, algorithm for biped locomotion is presented as the basis for control of an exoskeleton used to stabilize individuals with SCI. As has been established in robotic biped locomotion, foot placement is a critical component. Each step must be carefully planned based on feedback from the current robotic state space [11]. Choosing the next step carefully, the biped is shown to maintain its ZMP inside the support polygon as well as ensuring that the center of mass (COM) does not diverge from the ZMP [12]. This strategy, and those similar to it, depend on having a linear inverted pendulum model to find the desired ZMP and COM trajectories. In ideal situations, the inverted pendulum model can describe the real biped system well enough to predict the correct next step. However, our approach does not depend on having a known, fixed dynamics model. Instead, the model is obtained through a learning process where the data is used to train a neural network. The advantage of this approach is that it does not need any prior knowledge about the system. Furthermore, the use of a neural network is superior to a linearized dynamic model, as it can capture nonlinearities and make approximate or simplified models unnecessary [13].

In future work, the methods presented here will be applied to control the user’s muscles and powered lower limb exoskeleton based on an adaptable, reinforcement learning approach. To make the system robust for any user, the control approach must be adaptable [14]. It should thus function with limited a-priori information about the individual. To accomplish this, we employ an exploratory reinforcement learning type approach based on deep Q-networks (DQN) [15]. Reinforcement learning (RL) is a type of machine learning wherein a controller learns through trial and error. Over each trial and error episode, the controller is graded by a reward function that indicates how well it is performing [16]. The goal is to maximize the total reward, and thereby produce a controller that accomplishes some given task [17]. As control of a biped is defined over continuous state–action space, DQN is not directly applicable as it is natively applicable to discrete action space problems. A variation of DQN called deep deterministic policy gradient (DDPG) [18] is utilized here instead. Our system is composed of three separate controllers designed to operate together to produce stable walking control. One of the three controllers is a trained DDPG network and the other two consist of a conventional proportional–integral–derivative (PID) feedback controller and an open-loop controller. The use of three separate controllers reduces the complexity of the control system, and limits the number of training iterations required. As degrees-of-freedom increase, neural networks can have issues such as covariate shift [19] and increased training time. Since the application at hand is time sensitive, the speed of learning is crucial [20]. Furthermore, using three separate controllers allows for easy parallelization of the processes and dedicated multi-threading. We demonstrate the promise of the approach using a simulated biped and are encouraged by the results.

## 2. Methods

We trained a neural network via DDPG, which we refer to as the “DDPG network”. It works in conjunction with an open-loop controller and a PID feedback controller to induce stable walking locomotion. In this section, we first introduce our simplified bipedal model and our dynamic simulation thereof. Subsequently, we describe the target locomotion. Finally, we introduce our control approach for the model.

### 2.1. Biped Model

The biped model was developed in GAZEBO [21] and simulated using the Open Dynamic Engine [22]. The humanoid model consists of a torso and two legs and is based on anthropometric data, but constrained to move only in the sagittal plane. It was used in simulation to both train and test the effectiveness of the DDPG network. The model contains seven rigid bodies: the torso and the left and right thigh, shank, and foot. Additionally, the model has the following six joints: left and right hip, knees, and ankles. The hip and ankle joints can rotate along both the *x*-(medial–lateral) and *y*-(fore–aft) axes. Two frictionless walls are added in the simulation environment to constrain the biped to move in the sagittal plane, so the *x*-axis rotation of the ankle provides the majority of the movement. The *y*-axis rotation of the ankle is kept so the foot can make solid contact with the ground when the biped is tilting sideways. The knees are constrained to rotate about the *x*-axis only, giving the system a total of 10 degrees-of-freedom. The proportion of mass and length of the biped’s body segments are calculated from anthropometric tables based on the weight and height of the biped [23], while the shape and the rotary inertia of the bodies are simplified to a uniform box shape to speed up the simulations. In this work, unless otherwise noted, the height of the individual being modeled was set at 1.8 m with a weight of 75 kg. A simulated inertial measurement unit (IMU) sensor was attached to the center of the torso to measure its velocity and acceleration. This replicates what might be implemented on a powered exoskeleton. Force sensors are modeled on both the left and right feet to detect ground contact force. All joint angles and joint velocities can be directly read from the simulation environment.

### 2.2. Target Locomotion

Human gait is a complex process [24,25]. Here, the target gait is simplified into four phases for each leg: early swing, terminal swing, stance, and toe-off as shown in Figure 1.

#### 2.2.1. Early Swing Phase

In this phase, the thigh swings forward. The knee is flexed to prevent the swing foot from contacting the ground. The swing angle of the hip joint and the duration of the swing is determined by the output of the network trained via DDPG.

#### 2.2.2. Terminal Swing Phase

Following early swing, the hip joint is locked for a short duration allowing the knee to straighten. This move is in preparation for making ground contact.

#### 2.2.3. Stance Phase

Once the foot contacts the ground, the biped rotates around the ankle joint similar to an inverted pendulum. The hip joint is then unlocked. A PID controller is tuned to control the torso pitch via control of hip joint velocity.

#### 2.2.4. Toe-Off Phase

The stance phase of a given leg ends when the opposite foot makes contact with the ground and this opposite leg enters its stance phase. The end of the stance phase initiates the entrance into the toe-off phase. To do so, a torque is applied to the ankle joint to drive the foot to push off. This pushing action propels the biped forward. The amount of torque is determined by the desired walking speed. Following the pushing action, a torque is applied on the ankle joint to quickly flex the joint and retract the foot from the ground.

### 2.3. Control

In this section, we first introduce the control architecture and then subsequently describe each portion in greater detail.

#### 2.3.1. Controller Architecture

The controller architecture is outlined in Figure 2, which shows how the system switches between the three different controllers. The biped always steps with its right foot first. This is hard-coded for simplicity. Subsequently, the left leg of the biped enters its stance phase. During this phase, the stance controller is enabled for the left leg until the right foot contact sensor is activated due to right foot ground contact. Then, the right leg enters stance and the DDPG network outputs the step length and step duration for the left leg. The toe-off controller compensates for walking speed error before the left leg enters its swing phase.

#### 2.3.2. Deep Deterministic Policy Gradient

Here, the DDPG network is used to control the step length and step duration in the early swing phase. Deep deterministic policy gradient is a model-free policy learning method. It consists of an actor network that updates the policy parameters, and a critic network that estimates the action–value function. Deep deterministic policy gradient uses the expected gradient of the performance objective as a policy gradient instead of a stochastic policy gradient in order to estimate the correct gradient much more efficiently [18]. It was previously believed that the deterministic policy gradient of a model-free network did not exist, but later it was proved that it does indeed exist [18] and is easier to compute than stochastic policy gradients as it only needs to integrate in the state space.

The deterministic policy gradient is defined as follows:(1)$$\nabla_{\theta }J\left(\pi_{\theta }\right)=\int_{S}\rho^{\pi }\left(s\right)\nabla_{\theta }\pi_{\theta }\left(s\right)\nabla_{a}Q^{\pi }\left(s,a\right){|}_{a=\pi_{\theta }\left(s\right)}ds$$
where *J* is the performance objective, $\rho^{\pi }\left(s\right)$ is the state distribution, $\pi_{\theta }$ is the policy with parameters θ, and $Q^{\pi }\left(s,a\right)$ is the value of action *a* in state *s*, where $J\left(\pi_{\theta }\right)=E\left[r^{\gamma }|\theta \right]$ and $r^{\gamma }=\sum_{i=1}^{\infty }\gamma^{i}r\left(s_{i},a_{i}\right)$ is the cumulative discounted reward. The deterministic policy gradient can be treated as consisting of two parts. One part is the gradient of the action value with respect to the actions. The second part is the gradient of the policy with respect to the policy parameters.

The method of training and updating the network is iterative and depicted in Figure 3 and Figure 4. The DDPG network reads in the network state *s* from GAZEBO and the simulated sensors. This state is given to the actor network as input. The output of the actor network is the corresponding action *a*, which is the command to the actuator. This action moves the biped to its next state s_, which is evaluated and a reward *r* is given for s_. Meanwhile, the critic will give an action value q(s,a) corresponding to the state–action pair. A vector [s,a,r,s_], is stored in the memory buffer to use as training data. The sum of the future rewards is predicted by the target net. It takes the resulting states s_ as input to its actor net and gives a prediction of the next move a_ and the predicted next action value q_ is given by the critic in the target net. The action value is approximated by the critic using a neural network (NN). The parameters of the network are updated using a temporal-difference method in a similar way as traditional actor–critic methods. The actor also uses a NN as the policy. The policy parameters are updated by the deterministic policy gradient $\nabla_{\theta }J\left(\pi_{\theta }\right)$. A replay buffer is utilized to store transitions to break correlation in the sample trajectory. Learning directly from consecutive samples is inefficient, due to the strong correlations between the samples [15] and, when the actor network is trained, the policy will change constantly. Thus, the temporal difference is calculated by a copy of the actor–critic network which is called the target network. These networks only update after a period of time, or update with very small changes. This off-policy method allows the behavior to be more stochastic to explore the environment and keep the prediction deterministic. The target network is updated by a soft replacement method defined as follow:(2)$$\theta^{\prime }\leftarrow \tau \theta +\left(1-\tau \right)\theta^{\prime }$$
where θ is the network parameters (weight and bias array for neurons in the critic and actor networks), τ is the soft replacement factor, with constraint 1>τ>0. This replacement will ensure the target network updates slowly compared to the main net to solve the time correlation problem during training. The full biped system state includes position, velocity, and acceleration terms for all degrees-of-freedom. This large number of inputs can lead to network convergence issues and may require the use of a very large network to sufficiently understand the interactions of the many state variables. Therefore, measures were taken to simplify the input. Using simplified descriptors makes the learning more manageable, but requires domain knowledge to design an informative and compact feature set that may nevertheless be missing important information [26]. First, the biped was assumed to remain on the ground during normal walking; second, walking was limited to the sagittal plane by frictionless walls that prevent motion in the coronal plane; and third, a PID controller was used to stabilize the torso pitch angle. We reduced the model to include only the following: ϕ, torso pitch angle; *v*, torso forward speed; *l*, the actual step length; and $d_{zmp}$, the distance between the ZMP and the foot. The simplified input to the network (state *S*) is
(3)$$S=[v,\phi ,d_{zmp},l],$$
and the output of the DDPG network is
(4)A=[Len,t],
where *Len* is the step-length and *t* is the step duration. The PID that controls the torso pitch through the hip joint velocity is described in the next subsection.

The ZMP control is often used to control biped stability. If the ZMP is outside the support area, the biped can tip over and fall [27]. The state variables ϕ and *v* are measured by the IMU sensor attached to the torso. The step length *l* can be calculated from the forward kinematics in a physical system, whereas in simulation it can be read directly from GAZEBO. The ZMP is estimated after measuring the acceleration using the cart-table model [28]:(5)$$y_{zmp}=y_{com}-\frac{{\ddot{y}}_{com}}{g}z_{com},$$
where $y_{zmp}$ and $y_{com}$ represent the ZMP and center of mass of the biped on the *y*-axis (fore–aft), respectively. When the ZMP is obtained, the state parameter $d_{zmp}$ can be calculated by measuring the distance from the ZMP to the support foot. It should be noted that, in the simulation, the ZMP calculated using acceleration was relative to global coordinates, thus the location of the support foot was also measured in global coordinates. This is easy to implement in simulation; however, in the real world, this method is not easy to implement as the accuracy of the absolute position (global coordinates) of the ZMP or support foot obtained via IMU with global positioning system (GPS) may be too poor for control purposes. One solution is to use several pressure sensors under the foot to estimate the ZMP position. Although the biped is in a simulated environment, the acceleration measured by the IMU has noise due to the surface contact model in the physics engine, as shown on the top of Figure 5. Here, a mean value and Kalman filter was used on the acceleration data [29]. This is important because, in a human exoskeleton application, the measurements of the acceleration are often extremely noisy as well. The state is updated at the moment when the front foot contacts the ground, and then passed to the network, which returns an action. Decaying noise is added to the action chosen to promote initial exploration but then allows refinement over time.
(6)$$a^{\prime }∼\left(a,\sigma^{2}\right),$$
where $a^{\prime }$ is the action after adding noise, which is normally distributed with mean “*a*” (the original action) and standard deviation σ. Once training is completed, the system will run forward without additional noise added to the action selection.

The trained network decides how far and how fast to place the next swinging foot based on the velocity, torso pitch angle, step length and ZMP position of the previous step. The states are only sampled with every foot step. Consequently, if there is any major disturbance in between two foot steps, the network will not respond as fast as compared with other more quickly updated systems. The network must wait until the foot touches the ground to update the state. However, since the output of the network is the length and duration of the next step, as long as the biped does not fall between two steps, it can counter the disturbance by adjusting the output of the next step. To speed up the training, the output is initialized based on height-to-stride length ratio. A better starting point makes the network converge more quickly.

To train the control network using a reinforcement learning approach, a reward function is created to indicate if the actions taken by the controller are either good or bad. The reward function used here is based on the difference between the actual step length and the fixed reference step length.
(7)$$r=-{\left(l_{a}-l_{r}\right)}^{2},$$
where $l_{a}$ represent the actual step length and $l_{r}$ is the reference step length, $l_{r}$ = 0.75 m for the 1.8 m model. This reference step length is just a guess based on body height. A well trained DDPG network provides the action that maximizes the sum of the future rewards, while an over-trained network stubbornly tries to match the actual step length to the reference step length. To prevent overtraining, dropout layers [30] are added into the actor and critic networks. The training process is shown in Figure 6. The training is started when there are 20,000 transitions stored in the memory buffer, and after approximately 6000 steps, the biped can maintain continuous walking.

#### 2.3.3. Stance Controller

The DDPG network determines the desired step length and duration, and a separate controller, the stance controller, controls the biped during the stance phase. When the foot touches the ground, the biped starts to rotate around the ankle joint. In this phase, the hip joint needs to move according to the ankle joint to keep the torso up straight and provide power to drive the torso forward. The output of the stance controller is the target angular velocity of the hip joint. The goal is to keep the torso upright without overshoot because overshoot would cause the torso to pitch back and forth and jeopardize stability. Ideally, the torso is pitched slightly forward to maintain momentum and a smooth natural walking gait. To achieve this, a proportional stance controller was designed, and the residual error from this controller allows the torso to slightly pitch away from the *z*-(inferior–superior) axis.

With the torso pitch remaining constant with respect to the *z*-axis, the horizontal velocity of the hip will be the same as the horizontal velocity of the torso center,
(8)$$v_{t}=v_{p}$$
and
(9)$$\omega =-\dot{\alpha },$$
where $v_{t}$ and $v_{p}$ are the linear velocity of the torso center of mass and the hip joint, respectively, and ω is the angular velocity of the thigh about the hip joint as shown in Figure 7. The angular velocity about the ankle can be measured directly. The moment when the foot impacts the ground, noise is introduced. Thus, the angular velocity about the ankle is calculated by
(10)$$\dot{\alpha }=\frac{v_{p}}{\cos\alpha L}.$$

We thus designed a controller governed by the following equation:(11)$$\omega =K\phi =-\dot{\alpha },$$
where, if the torso pitch angle ϕ is larger than the target value $\phi_{0}$:(12)$$\phi >\phi_{0},$$
then
(13)$$\left|\omega |>\right|\dot{\alpha }|.$$

Thus, the pitch angle decreases and vice versa. The control gain *K* is
(14)$$K=\frac{-\dot{\alpha }}{\phi_{0}}=\frac{-v_{p}}{\cos\alpha L\phi_{0}},$$
where target pitch is chosen to be close to zero:(15)$$\phi_{0}=0.02.$$

The target pitch cannot be set too close to zero, otherwise it produces a very large gain, causing the system to be sensitive to noise.

#### 2.3.4. Ankle Torque Control

The ankle joint is passive except in the toe-off phase. The advantages of setting the ankle to be passive are as follows: (1) smoother ground contact for the foot; (2) the dynamic property of the inverted pendulum is maintained; (3) minimal force is needed to drive the biped around the ankle when the foot is in contact with the ground; and (4) total noise in the system is reduced. The damping coefficient of the ankle was set to 1. This amount of damping helps to absorb the impact from ground contact without hindering the swing motion.

In the toe-off phase (Figure 8), a torque is applied on the ankle to propel the biped forward. The torque is determined by the current walking speed. The goal is to maintain the momentum of the biped within a certain range. If the desired walking speed is given, then
(16)$$\Delta v=v_{0}-v_{desire},$$
where $v_{0}$ is the current torso velocity, and, if the torso pitch remains constant, then the angular velocity of the torso is zero:(17)$$\omega_{torso}=0.$$

Subsequently, the velocity of the hip is equivalent to the velocity of the center of the torso,
(18)$$\Delta v_{center}=\Delta v_{hip}.$$

Assuming the toe-off phase is short, the hip joint angle of the rear leg remains approximately the same during the toe lift off, and the momentum of the rear foot can be neglected. To keep the torso angular velocity $\omega_{torso}=0$, a torque $\tau_{hip}$ must act on the hip joint of the front leg:(19)$$\tau_{hip}\Delta t=J_{torso}\Delta \dot{\alpha },$$
where $\tau_{hip}$ is the torque acting on the hip joint and $J_{torso}$ is the moment of inertia of the torso:(20)$$J_{torso}\approx \frac{1}{3}mh^{2}.$$

For the front foot ankle joint, we have the following:(21)$$\left(\tau -\tau_{c}-\tau_{hip}\right)\Delta t=J_{leg}\Delta \dot{\alpha },$$
where τ is the torque acting on the ankle joint, $\tau_{c}$ is the torque caused by the damper, and α is the angle between the leg and the vertical axis. We can express $\Delta \dot{\alpha }$ as
(22)$$\Delta \dot{\alpha }=\frac{\Delta v_{hip}/\cos\alpha }{l},$$
where $J_{leg}$ is the moment of inertia of the front and rear legs about the front ankle joint:(23)$$J_{leg}\approx \frac{1}{12}m_{l}l^{2}+m_{l}\left[l^{2}sin^{2}\beta +{\left(l\cos\beta -\frac{l}{2}\right)}^{2}\right]+\frac{1}{3}m_{l}l^{2}.$$

We express $\tau_{c}$ as follows:(24)$$\tau_{c}=c\dot{\alpha },$$
where *c* is the damping coefficient of the ankle joint.

### 2.4. Simulation and Testing

The biped was simulated using GAZEBO and controlled through ROS [31]. GAZEBO is an open source simulator, while ROS is a set of software libraries and convenient tools used for robotic systems [32]. Joint movement in the GAZEBO simulation is controlled in two ways. First, we can call the ROS function "ApplyJointEffort" directly to set a torque value for some duration. Secondly, we can also use GAZEBO’s controller plug-in. The controller plug-in provides three different PID control methods: torque feedback, velocity feedback, and position feedback. Here, the plug-in’s velocity feedback control was utilized. The PID parameters were tuned to react in a fast and stable manner.

## 3. Results and Discussion

To test the stability of the walking biped in simulation, impulses were applied to the torso (Appendix A). In simulation, the biped was first set to walk at a desired walking speed. We set this speed to approximately 1 m/s, which is the target speed for an individual wearing an exoskeleton in community ambulation. During testing, all impulses were applied for a duration of 0.1 s. The placement and magnitude of the impulses were varied.

In simulation, it was found that the biped was able to remain stable and continue walking after a maximum impulse of 30 kg·m/s was applied to the back of the torso in the direction of walking as well as after a maximum impulse of 40 kg·m/s was applied to the front of the torso, opposing motion. The typical result of these impulses is shown in Figure 9. After applying the impulse, the biped’s velocity increased or decreased, depending on the direction of the impulse, but then returned to a consistent oscillation in less than 5 s. The peak-to-peak amplitude of the torso oscillations was constrained to about 0.3 rad (Figure 10). It was found experimentally that the biped was able to resist larger disturbances when it was in the toe-off phase of the gait compared to the early swing phase. Increasing the target walking speed $v_{desire}$ and lowering the damping coefficient of the ankle joint were both found to increase the overall speed of the biped but reduce the robustness of the system.

Figure 11 shows that a positive impulse disturbance (in the walking direction) applied to the torso caused an increase in torso speed. To recover from this disturbance, the DDPG network increased step length and decreased step duration accordingly to regain stability. When a negative impulse was applied to the biped, the DDPG network reduced step length and increased step duration to adapt to a lower speed. All the adjustments made by the DDPG network to retain stability were learned purely by experience without prior knowledge. The simplified input state proved to be sufficient to train a successful network.

An even further simplified state input $s=[d_{zmp},\phi ]$ was also used to train a network with the same parameters, but even after an extended training period, it did not converge. This oversimplified state input cannot describe the system adequately, thus the DDPG network could not make the right decisions.

### Robustness to Different Body Sizes and Payloads on the Torso

The size and weight of the biped model was varied to further test the robustness of the control system. The biped model used in the simulation was built based on anthropometric data. The segment lengths were proportional to model height and segment masses were proportional to total body mass. The total body mass was roughly estimated by M=h−105 for a normal build where *h* is the body height in centimeters, and *M* is the body mass in kg. The DDPG network was trained for a male 1.8 m tall and average build. We varied the model height and associated anthropometric segment lengths and masses while keeping the controller the same to further evaluate its robustness.

For this test, the displacement of left/right foot and ZMP in the *y* direction (walking direction) was recorded to track the walking stability. The height of the model was altered from its original height by 0.1 m in each trial. The results show that the simulated biped remained stable, steadily walking when the height was no less than 1.4 m (Figure 12). When the height was set to 1.3 m, the biped stopped walking eventually (Figure 13).

In terms of disturbances, we found a smaller simulated biped could resist smaller disturbances. When the height was set to 1.4 m, the simulated model could only resist 20 kg·m/s of impact in the walking direction. This was expected given the decrease in body mass.

When the height and weight of the simulated biped were increased, the control system continued to function for a biped as tall as 2.3 m (Figure 14). It failed when the height was larger than 2.3 m. Tests also showed that, when the foot became too large, it started to contact the ground during swing, causing the biped to tip over (Figure 15).

When the body mass was increased, the biped could resist even larger disturbances. When its height was between 2.1 and 2.2 m, the biped could resist 60 kg·m/s impact load in the direction of walking. This was expected because of the increase of the body mass.

In addition, to test the robustness of the control system further, a load was added on the back or front of the torso (Figure 16). The results show that the biped based upon a 1.8 m tall male could carry up to 25 kg when the load was added on the front and 10 kg when the load was added on the back. When a 30 kg load was added on the front, the biped accelerated forward to compensate the torque introduced by the payload. When the walking accelerated to too great of a speed, the biped fell (Figure 17). When a 10 kg load was added on the back, the walking speed decreased by about half. As shown in Figure 18, during normal walking the biped would take up to eight steps vs. four steps in the same amount of time with a 10 kg backpack. The control system could not maintain walking when 15 kg was added on the back. Figure 19 is a schematic that summarizes the robustness of the controller for changes in body size.

Although the DDPG network decreased step frequency and step length to match the slow walking speed, the toe-off controller failed to compensate for the velocity error because it was designed considering body mass alone and not for the extra payload. The mass distribution of the model in the simulation was based on a human. However, the additional mass of the exoskeleton would change this. Therefore, we performed experiments with mass added to the legs to represent the mass of the exoskeleton. The walking stability was evaluated by the distance between the ZMP and the support foot. The result is shown in Figure 20. Ten kilograms or less had little influence on the biped’s stability. When the added mass became larger, the ZMP transition time between the two feet (large peaks in left figure) grew longer. Walking speed decreased when mass was added. In addition, the reaction force from the motor grew larger when swinging a leg, which caused increasing torso pitching angle.

## 4. Conclusions

We are developing a walking system for people with SCI. This walking system consists of a human with neural implants for muscle stimulation and an exoskeleton with motors for assisting the human’s muscles on an as-needed basis. Our goal is for the person to be able to walk for extended periods of time at speeds as fast as 1 m/s, community ambulation, stabilized primarily by the exoskeleton. The research described here developed and demonstrated a controller for walking stability in the sagittal plane for people of various heights, despite static and dynamic perturbations.

A biped model, based on anthropometric data, was implemented in the GAZEBO simulation environment for walking in the sagittal plane and a novel controller was developed for it. The ideal foot placement to maintain stability during walking was predicted using a DDPG neural network that was trained in simulation. The complexity of the network was decreased through a down-select of the state variables. The walking controller also included a PID system for the hip joints during their respective stance phases and an open-loop toe-off controller at the ankle joint.

The controller was shown to stabilize the biped despite static and dynamic perturbations. The simulated biped walked at a steady pace of approximately 1 m/s, and during locomotion, it maintained stability despite a 30 kg·m/s impulse applied forward on the torso or a 40 kg·m/s impulse applied rearward. It also maintained stable walking with a 10 kg backpack or a 25 kg front pack.

It was shown that the controller functions for people of different sizes without having to be retrained. The DDPG network was trained for a male 1.8 m tall and average build. We varied the model height and associated anthropometric segment lengths and masses while keeping the controller the same to further evaluate its robustness. The controller was shown to stabilize models 1.4–2.3 m tall. As expected, the resistance to dynamic loads was found to be proportional to body mass: a larger, more massive model could resist larger disturbances.

The robustness of the controller to a change in mass distribution to mimic the addition of the exoskeleton was also explored. It was shown that adding 10 kg or less to each leg has little influence on the biped’s stability without retraining the controller. In future work, when the exoskeleton is completed, its actual mass properties will be added to the model and the stability of the system will be tested. The DDPG controller will be retrained and the system will be tested again for comparison. Other future work may include real-time learning to improve stability with increasing static loads or for different individuals with a great difference in mass and mass distribution.

## Figures and Tables

**Figure 1 biomimetics-04-00028-f001:**
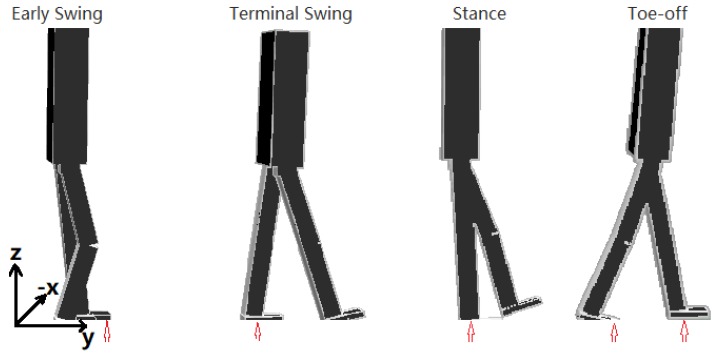
Simplified gait cycle for the right leg. Red arrows indicate which foot is contacting the ground.

**Figure 2 biomimetics-04-00028-f002:**
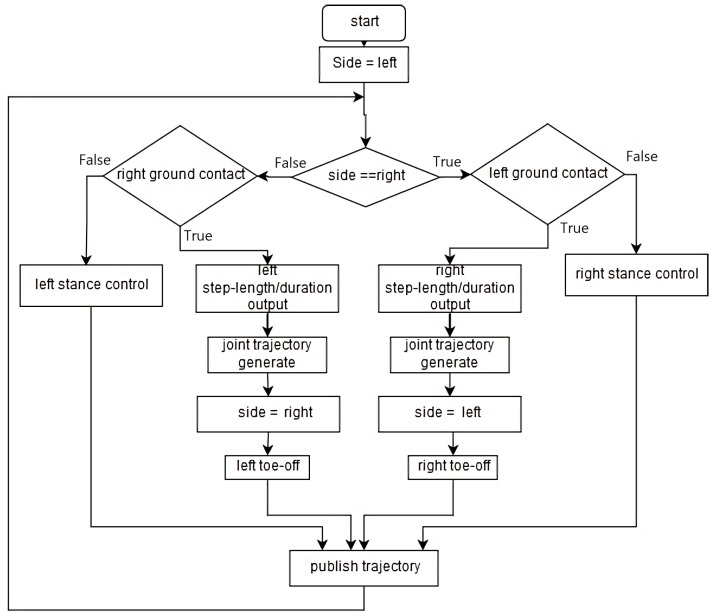
Flow diagram of the control system structure.

**Figure 3 biomimetics-04-00028-f003:**
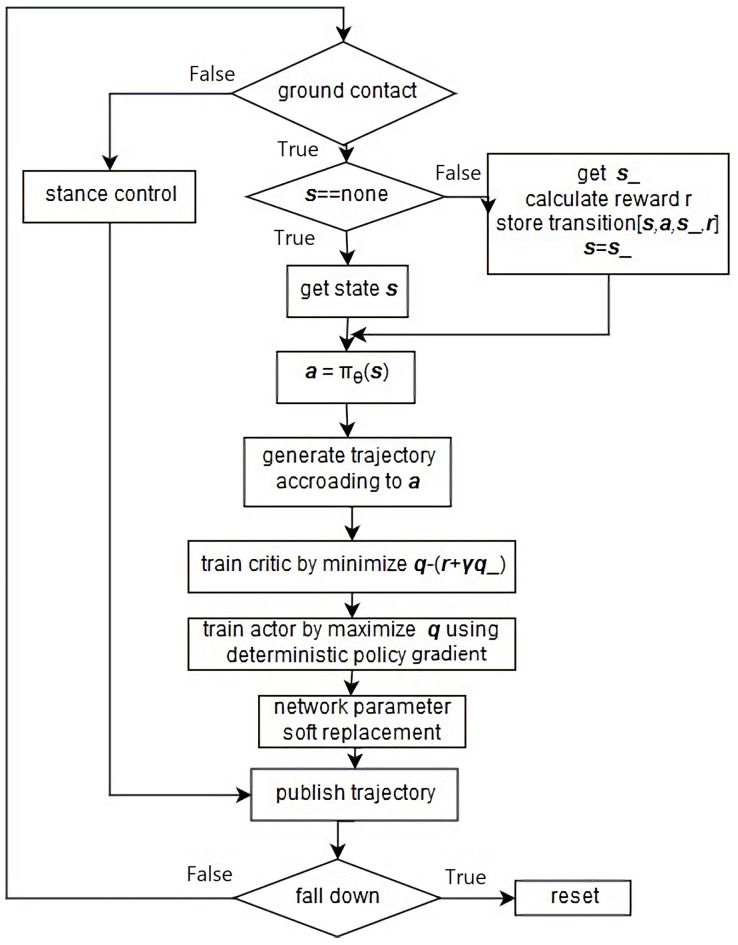
Flow diagram of DDPG network training process.

**Figure 4 biomimetics-04-00028-f004:**
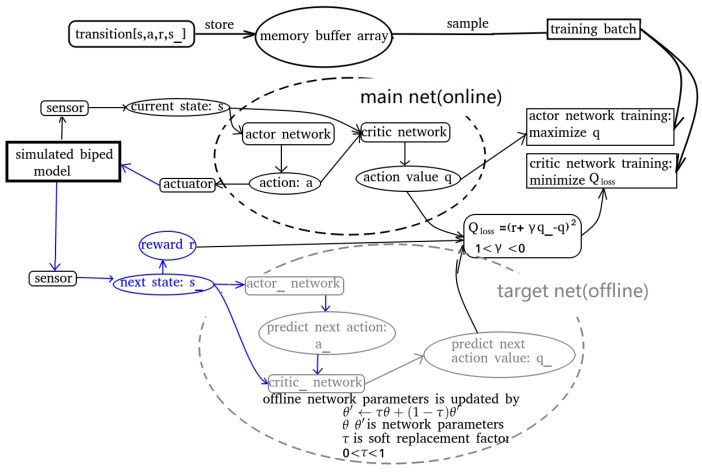
The critic and actor networks (evaluate/target) both have two hidden layers; the first layer has 400 neurons and the second layer has 300 neurons. The activation function is a rectified linear unit (ReLU). The output of the actor network goes through a tanh activation function. The network has a memory of 70,000 steps and the training begins when 20,000 transitions are stored. The learning rate of the actor and critic are set to 10${}^{-8}$ and 2 × 10${}^{-8}$, respectively. The reward discount γ is set to 0.99, and the training batch is 100 samples.

**Figure 5 biomimetics-04-00028-f005:**
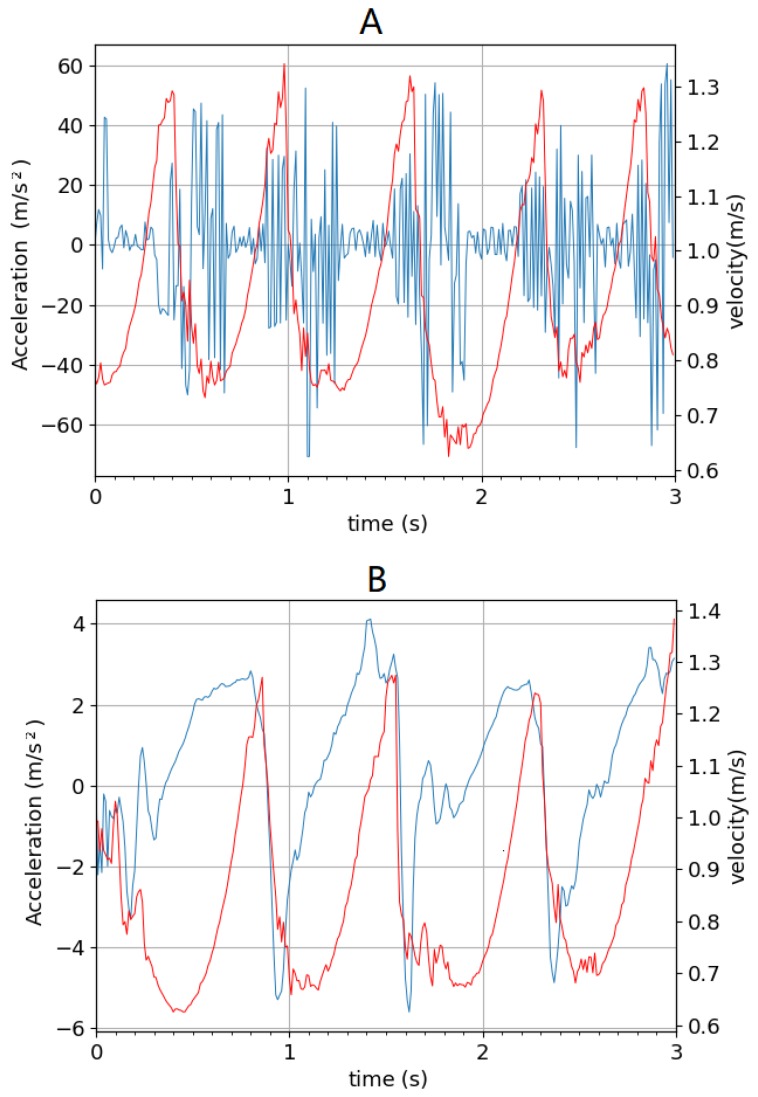
(**A**) Before and (**B**) after filtering. The blue line indicates the acceleration; the red line indicates the velocity.

**Figure 6 biomimetics-04-00028-f006:**
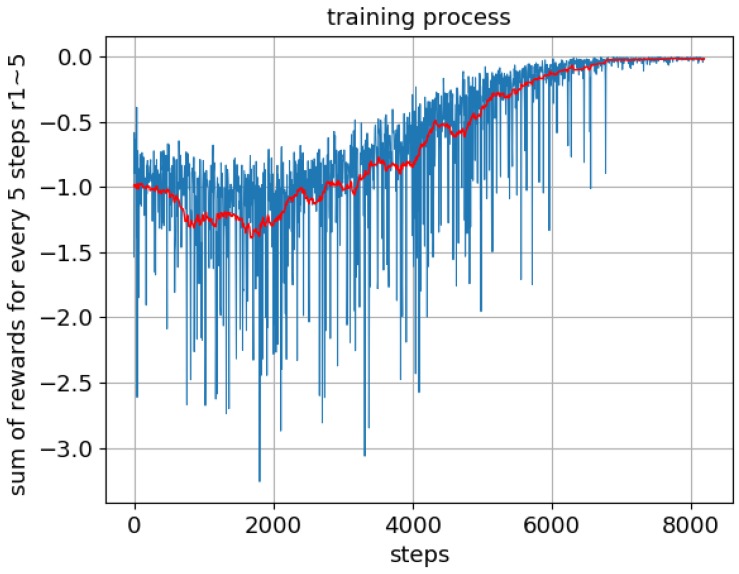
Convergence of the training process. The sum of the rewards versus the number of steps.

**Figure 7 biomimetics-04-00028-f007:**
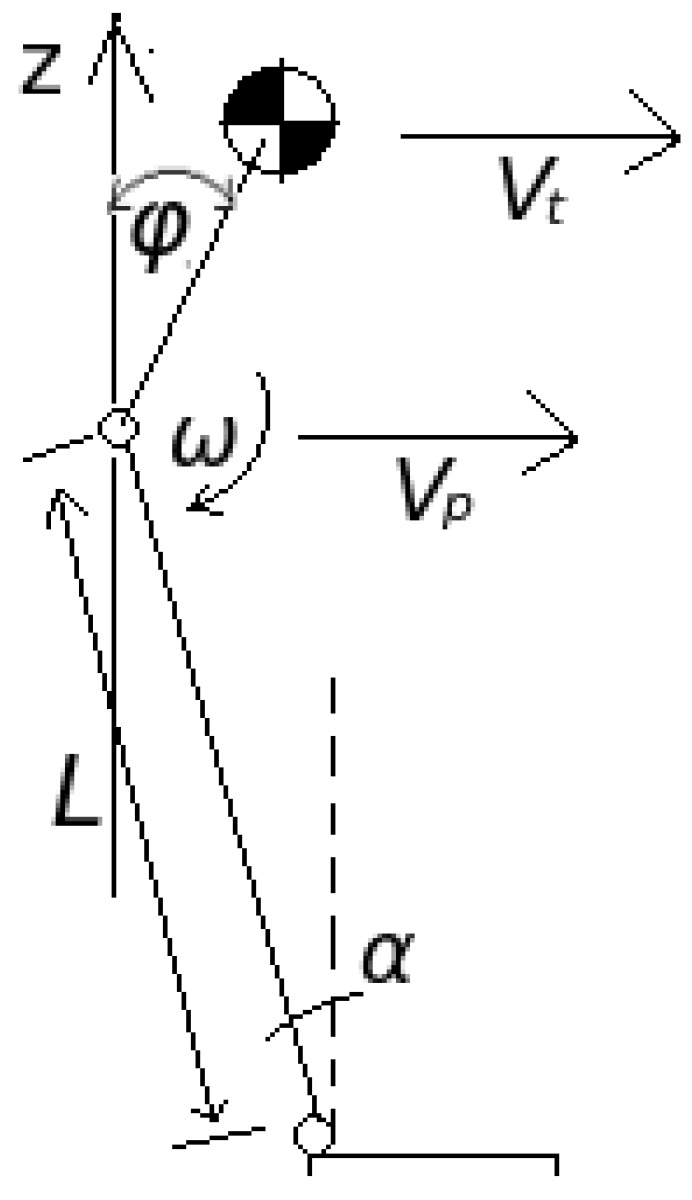
Model of leg and torso.

**Figure 8 biomimetics-04-00028-f008:**
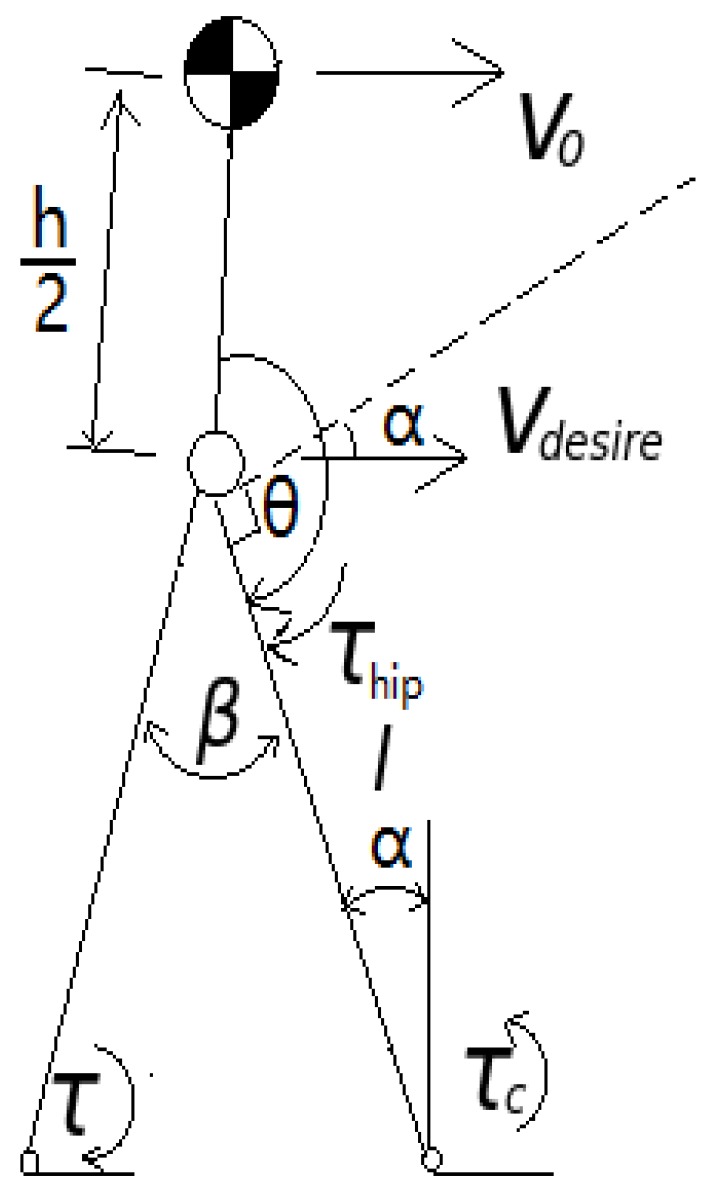
Model of biped with torques applied about ankle and torso.

**Figure 9 biomimetics-04-00028-f009:**
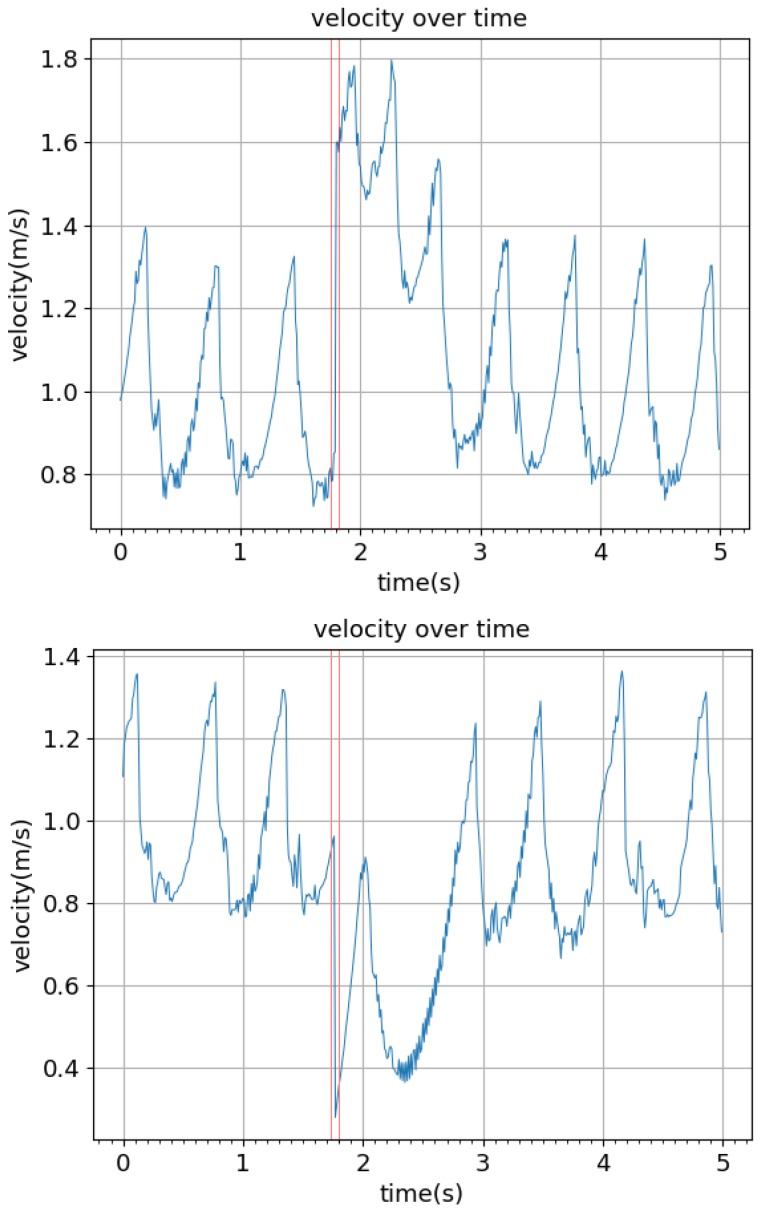
Impact of outside disturbances on the walking speed. Walking speed vs. time is plotted. (**top**) A 30 kg·m/s impulse is applied to the torso in the walking direction. (**bottom**) A 40 kg·m/s impulse is applied to the torso opposite to the walking direction.

**Figure 10 biomimetics-04-00028-f010:**
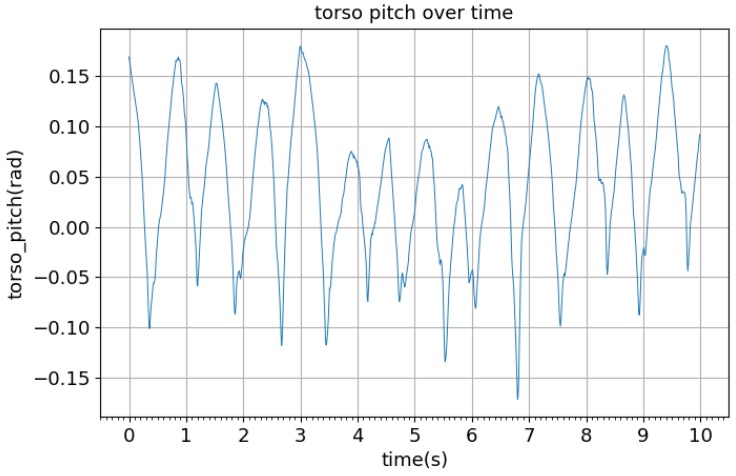
Torso pitch angle during walking.

**Figure 11 biomimetics-04-00028-f011:**
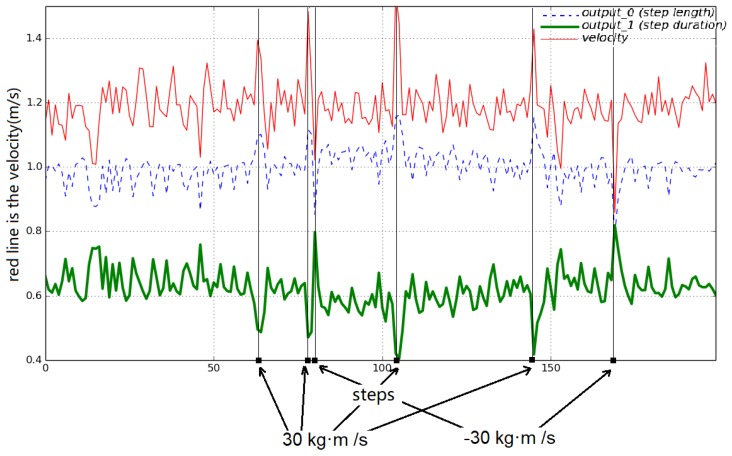
Deep deterministic policy gradient (DDPG) output over different speeds. The red line is the speed, and the green and blue lines are DDPG outputs, which are scaled and biased so that they can fit in the same graph for comparison.

**Figure 12 biomimetics-04-00028-f012:**
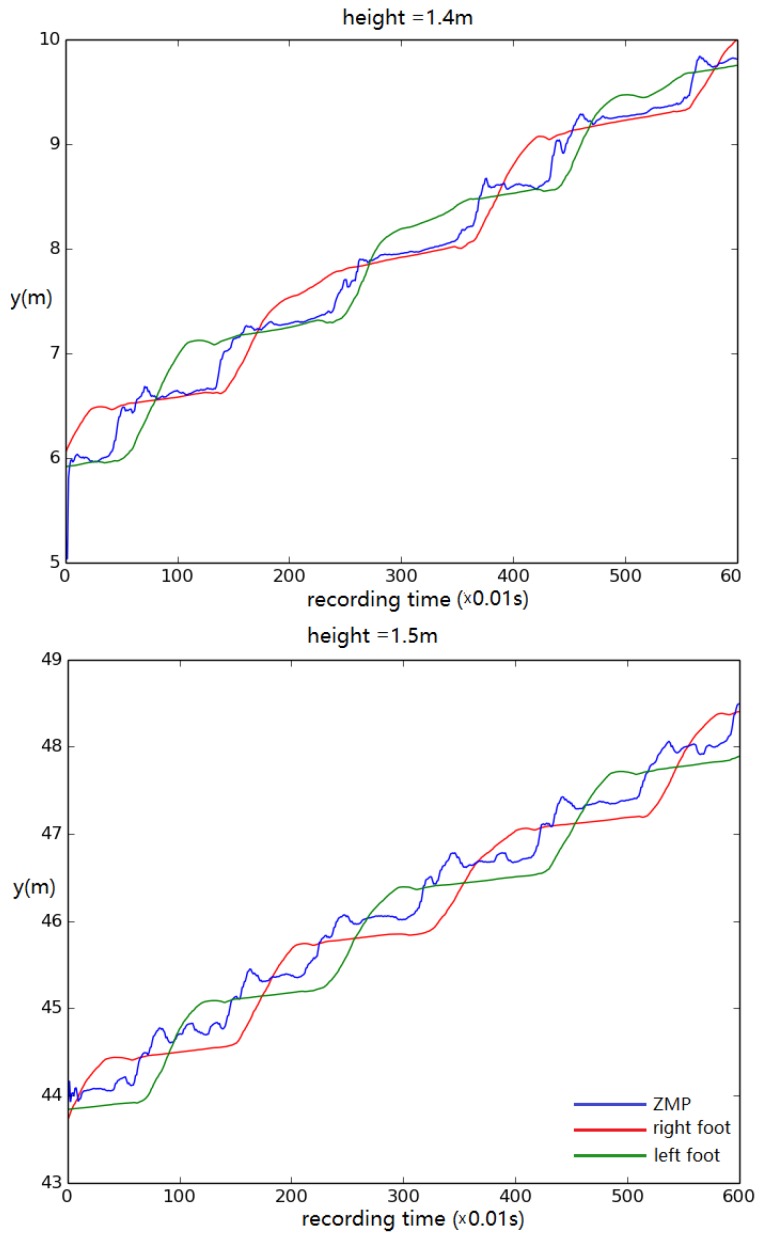
Foot and ZMP displacement when the body height is set to (**top**) 1.4 m and (**bottom**) 1.5 m. Displacement of torso center of mass vs. time.

**Figure 13 biomimetics-04-00028-f013:**
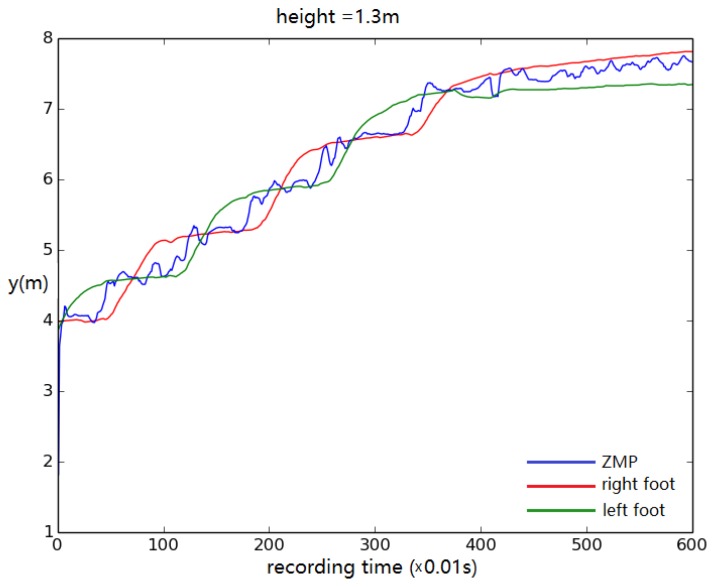
Foot and ZMP displacement when the body height is set to 1.3 m. Displacement of torso center of mass vs. time.

**Figure 14 biomimetics-04-00028-f014:**
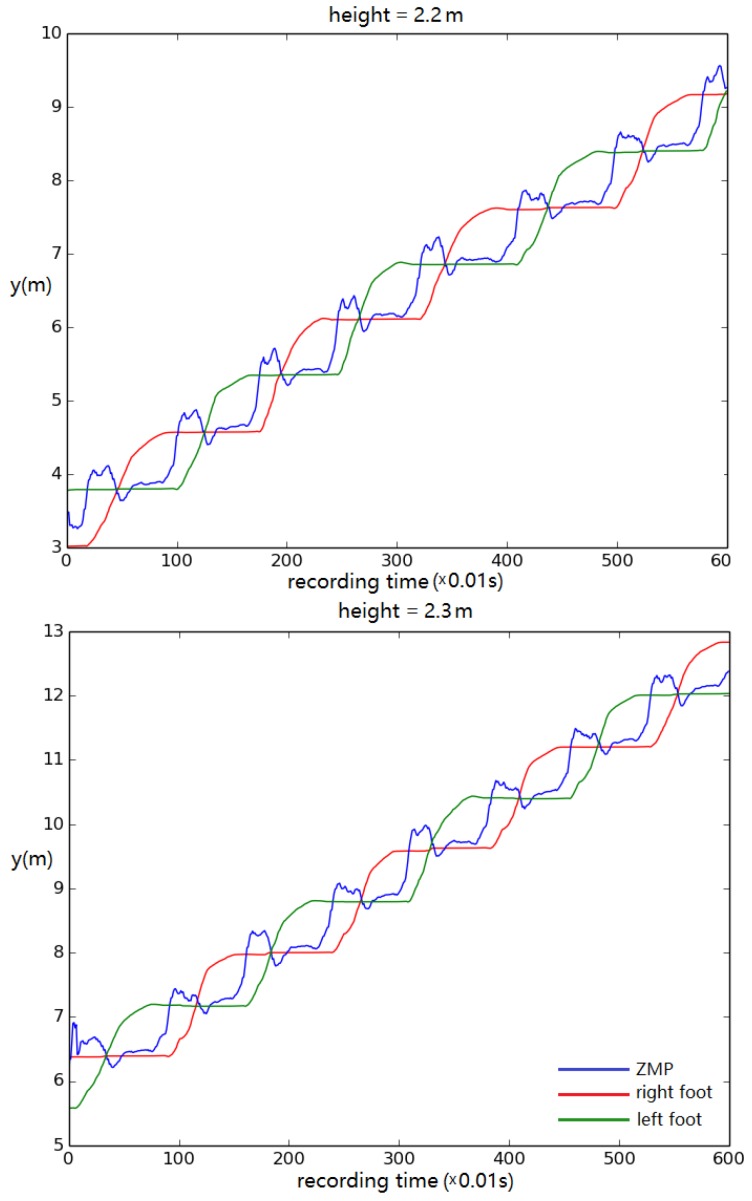
Foot and ZMP displacement when the body height is set to (**top**) 2.2 m and (**bottom**) 2.3 m. Displacement of torso center of mass vs. time.

**Figure 15 biomimetics-04-00028-f015:**
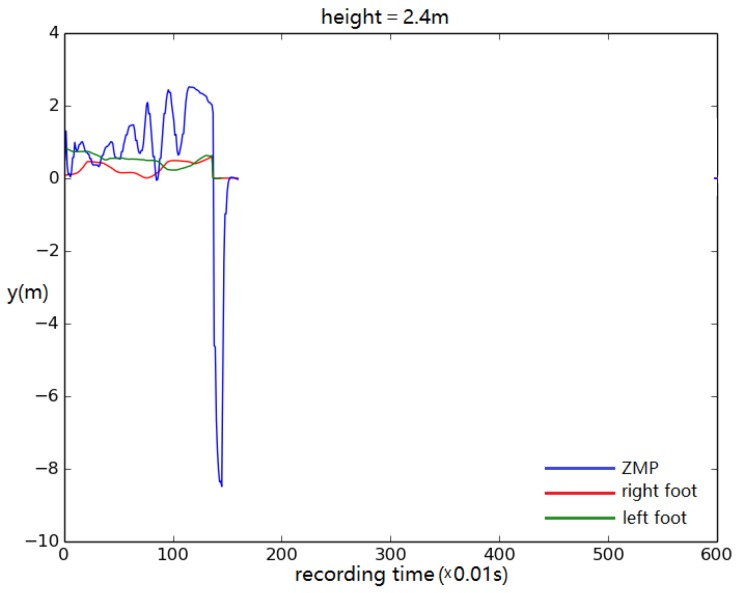
Foot and ZMP displacement when the body height is set to 2.4 m. Displacement of torso center of mass vs. time.

**Figure 16 biomimetics-04-00028-f016:**
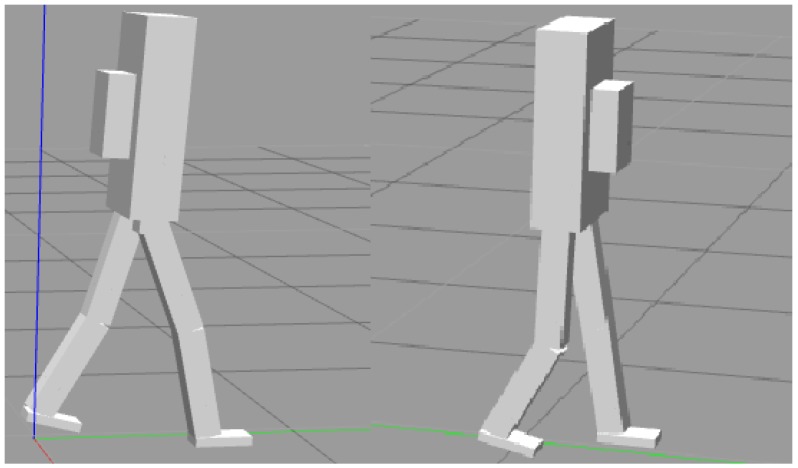
A simulated pack is added on the torso. Backpack on the left and front pack on the right.

**Figure 17 biomimetics-04-00028-f017:**
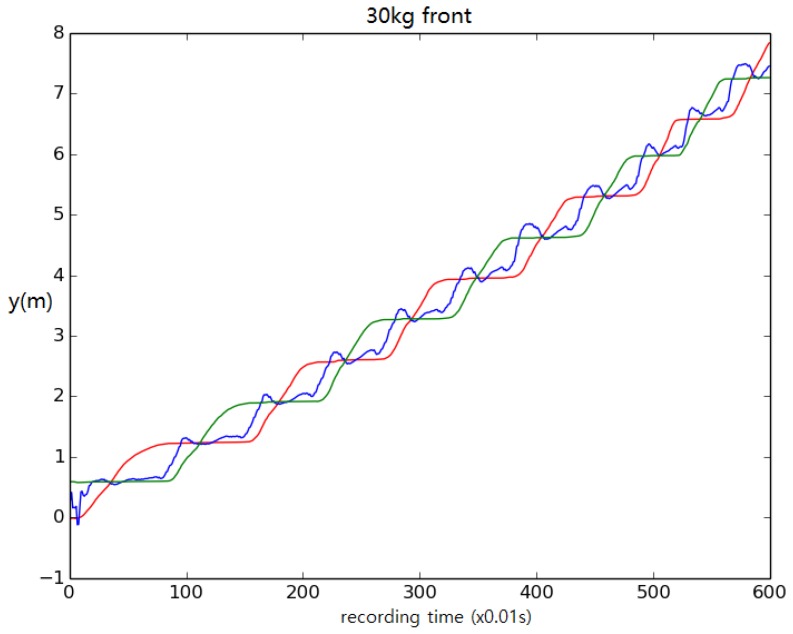
A 30 kg pack added on the front of the torso causes the walking speed to increase. The duration of each step gets shorter and the length of the stride increases. Walking distance versus time.

**Figure 18 biomimetics-04-00028-f018:**
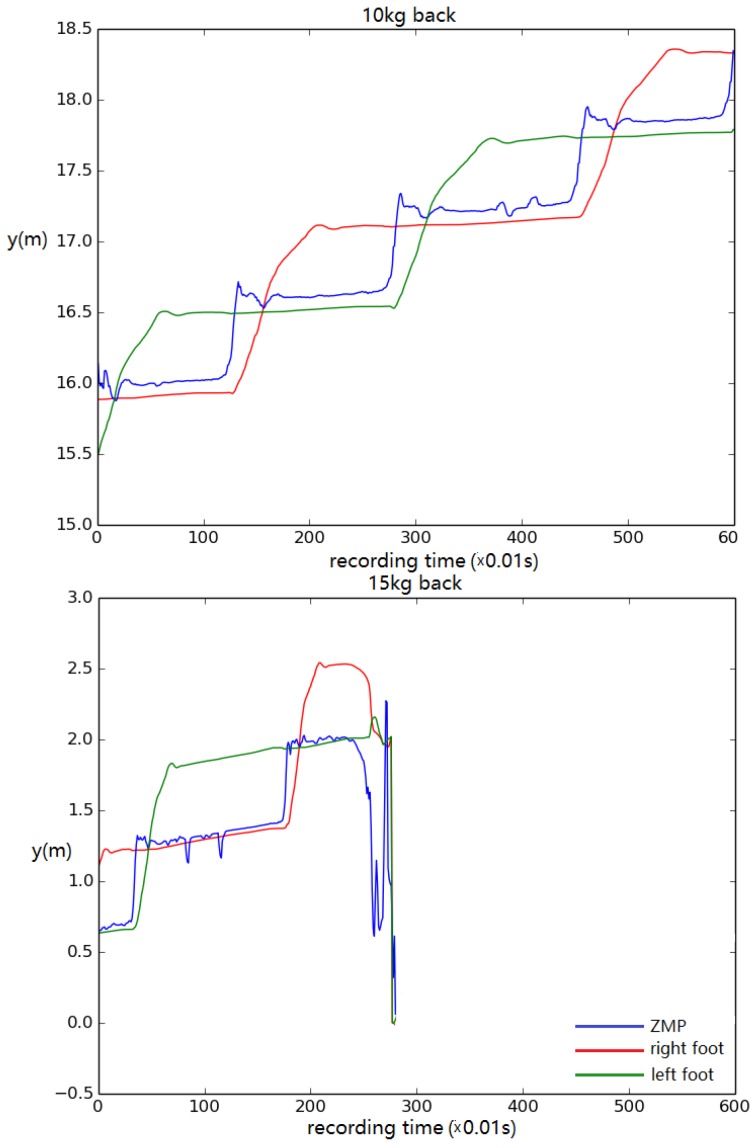
The simulation (**top**) walks with a 10 kg backpack with no change to the controller, but (**bottom**) falls with a 15 kg backpack. Walking distance versus time.

**Figure 19 biomimetics-04-00028-f019:**
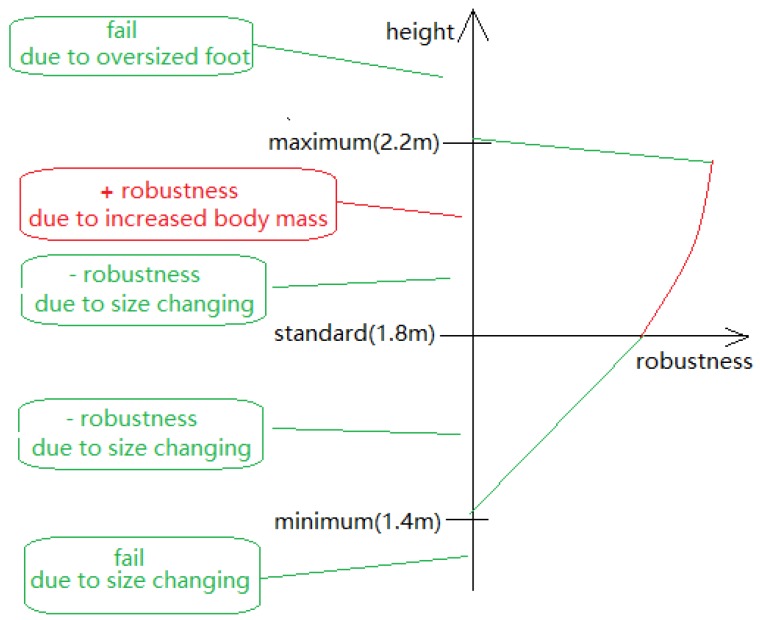
A summary of how the controller robustness changes with body size.

**Figure 20 biomimetics-04-00028-f020:**
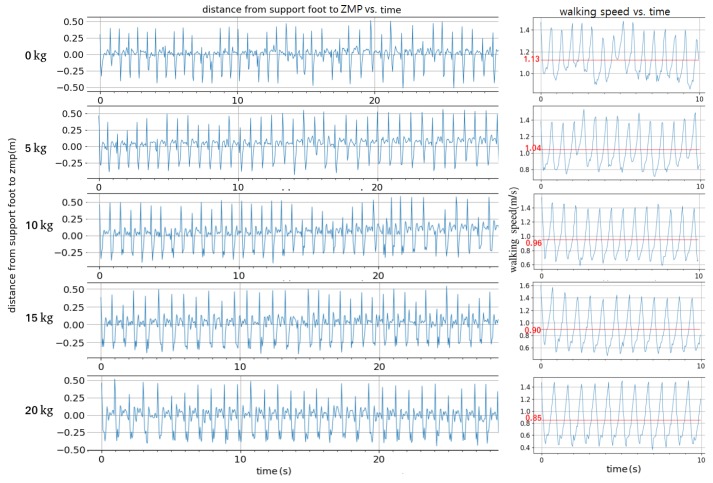
Effect on ZMP and walking speed from adding mass to the legs vs. time.

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
