# Peer review of "Implementation of Deep Deterministic Policy Gradients for Controlling Dynamic Bipedal Walking"

_biomimetics, 2019, doi:10.3390/biomimetics4010028_

Reviewer 1 Report

The paper presents a sufficiently complex bipedal robot model (sagittal plane only). An open loop controller for the stance, hip, and ankle are assumed and a PID controller is used for stabilization of the open loop controller. A Deep Deterministic Policy Gradient network is used to tune the parameters of the controller. Intuitive decisions are made on how to reduce the parameters set, a standard practice when the system is complex. The ensuring controller is shown to be robust to parameter variations and pushes.

There are multiple issues with the paper

1) The literature review is very limited. There have been multiple papers on usign Reinforcement Learning (RL) and Deep Learning (DL) algorithms that have not been included. For example, See UBC Michiel van de Panne, CMU Atkeson, MIT Tedrake etc.

2) What is novel in this work? It seems that DDPG might not have been used for locomotion. But it is not clear why DDPG is particularly better than other RL, DL algorithms.

3) Why do all graphs look noisy? I was expecting to see periodic gaits that have periodic profiles

4) The discussion is missing (even though it is in the title of section 3). The discussion section should talk about how these results compare to other works and/or broad generalization if any.

The paper ends abruptly.

5) In the abstract, the paper changes from past tense to present tense.

6) An animation video showng the results would help substantially. 

Author Response

Thank you very much for the comments. It is very useful for us to improve our manuscript. Below are the changes we made (in red font) according to your comments (in black font). The primary changes in the manuscript are also marked in red font. However, many minor grammatical changes are not marked.

Reviewer 1

1.       The literature review is very limited. There have been multiple papers on using Reinforcement Learning (RL) and Deep Learning (DL) algorithms that have not been included. For example, See UBC Michiel van de Panne, CMU Atkeson, MIT Tedrake etc.

We agree that the introduction and the minimal citations to previous work were major weaknesses. The introduction section is substantially rewritten and extended. The number of referenced papers is tripled and include many more citations of RL and DL algorithms including by Panne, Atkeson and Tedrake.

2.     What is novel in this work? It seems that DDPG might not have been used for locomotion. But it is not clear why DDPG is particularly better than other RL, DL algorithms.

Continuous state space and action space limits the use of other RL/DL algorithms. The DDPG uses an actor-critic structure that can handle such problems and it is more efficient than using a stochastic policy gradient. But, the actor-critic structure requires training two networks that interact with each other at the same time, and so limiting the size and complexity of the DDPG network is important. This work provides a method for implementing DDPG into a locomotion control system by simplifying the state variables and using auxiliary controllers

3.     Why do all graphs look noisy? I was expecting to see periodic gaits that have periodic profiles

Disturbances are added at about 2 sec in figure 9, but periodic profiles still can be seen.

The scale for the y axis in fig 10 is small, so the difference between two periods looks large.

The sample frequency in figure 11 is per foot-step, so the curve is not periodic.

4.     An animation video showing the results would help substantially.

We have provided a screen recorded video.

Reviewer 2

1.       The objective of the paper is not defined well. It is argued that the main aim is to develop a controller for the exoskeleton. The developed controller has no association with the human walking control. The authors developed a nice controller that is able to stabilize the walking of a robot but there is no evidence that this controller can be used on an exoskeleton. If the controller has to be applied to Individuals with spinal cord injury (SCI), several validation experiments have to be delivered to compare the behavior of the controller and human. The main question is that if the controller can adapt itself to human body dynamics. If the controller is developed for more broad robotic applications, there should be a validation experiment regarding the comparison between this method and other walking controllers. The paper is descriptive of the proposed methodology where it fails to motivate why the robotic community should adopt yet another controller for biped robots.

The introduction and discussion sections are substantially rewritten and extended to address this point. The final goal for this controller is for a human/exoskeleton system. However, at this level of complexity, there is little difference between biped robot control and human/exo walking control if they have the same mechanical designs. This paper shows an implementation of deep reinforcement learning for biped locomotion control. Now that it is shown that DRL can handle such problems, a dedicated network can be trained for a specific system, human/exo or robot. The dynamics of the two systems will differ from each other (e.g. the control of human walking may need to transform the motor torques into muscle activations) but generally DNN treats it as a black box. New results are added to show that the system trained for a human of a particular size and mass also works when additional mass is added to the legs, representing the mass of the lower-limb exoskeleton. However, in future work, when the exoskeleton is finalized, the system will be retrained with the human/exo inertia properties.

2.     The only property of interest of the methodology is introduced as geometry scalability of the biped model and the controller is invariant to the changes in the body size.
Where this problem can be easily addressed by building a tuning table of the controller parameters for different morphologies.

The fact that the “controller is invariant to changes in the body size (and mass)” shows the stability margin of the controller. A tuning table would be sparse for this control system. In practice, we plan to use this controller as a starting point and retrain it for models of different human subjects (wearing the exo) to increase the stability of the system even further for a particular person. It is good to know that the results in this paper show that the system remains stable despite large changes in static loads and even for large impact forces, for example, when the person dons a backpack, picks up bag of groceries or is inadvertently pushed by another person. This is very important for a person/exo system in the real world.

3.     Depending on the objective of the paper the introduction section fails to provide a broad overview of the literature in this domain. Please extend the introduction part

The introduction section is substantially rewritten and extended. The number of referenced papers has been tripled.

4.     The main content of the paper is a description of the controller architecture which necessitates an addition of the block diagram for the proposed method.

Block diagrams have been added as figure 2 and figure 3, the Controller Architecture is explained in section 2.4. We rearranged the paper so the order of the figures has changed.

5.     Please improve figure 2

Figure 2 (now figure 4) is redrawn and the text describing it has been clarified.

6.     The main concern about the content of the paper is that it is written as a stand-alone paper where few citations have provided supporting the design decisions of the controller.

Again, we agree that this was a major weakness of the paper. The introduction was rewritten and the number of citations was tripled to set this work in its proper place within the literature.

Reviewer 2 Report

This paper explains flawlessly the proposed methodology for bipedal robot control in the Sagittal plane.The control is simplified into three parts which are a combination of the learning methods for the body posture hyper-parameters and PID controller based on the ZMP model and the inverted pendulum.The authors comprehensively describe the method and its specific parts however the paper still has several major shortcomings as a journal paper.

*) The objective of the paper is not defined well. It is argued that the main aim is to develop a controller for the exoskeleton.
The developed controller has no association with the human walking control.
The authors developed a nice controller that is able to stabilize the walking of a robot but there is no evidence that this controller can be used on an exoskeleton.
The only property of interest of the methodology is introduced as geometry scalability of the biped model and the controller is invariant to the changes in the body size.
Where this problem can be easily addressed by building a tuning table of the controller parameters for different morphologies.
If the controller has to be applied to Individuals with spinal cord injury (SCI), several validation experiments have to be delivered to compare the behaviour of the controller and human.
The main question is that if the controller can adapt itself to human body dynamics.

If the controller is developed for more broad robotic applications there should be a validation experiment regarding the comparison between this method and other walking controllers.
The paper is descriptive of the proposed methodology where it fails to motivate why the robotic community should adopt yet another controller for biped robots.

*) Depending on the objective of the paper the introduction section fails to provide a broad overview of the literature in this domain.
Please extend the introduction part

*) The main content of the paper is a description of the controller architecture which necessitates an addition of the block diagram for the proposed method.

*) Please improve figure 2.

*) It looks like the DDPG controller is very restricted in performance and learning, It is recommended that authors add a section regarding the DDPG network analysis its pros and cons.

*) The main concern about the content of the paper is that it is written as a stand-alone paper where few citations have provided supporting the design decisions of the controller.

In summary, the paper is well written and the proposed controller is also promising; however, the authors have to provide better evaluations of their work with respect to a specified reference.
This reference could be other bipedal controllers or human. This reference should be elaborated in the introduction and motivate the adoption of the proposed method.

Author Response

Thank you very much for the comments. It is very useful for us to improve our manuscript. Below are the changes we made (in red font) according to your comments (in black font). The primary changes in the manuscript are also marked in red font. However, many minor grammatical changes are not marked.

Reviewer 1

1.       The literature review is very limited. There have been multiple papers on using Reinforcement Learning (RL) and Deep Learning (DL) algorithms that have not been included. For example, See UBC Michiel van de Panne, CMU Atkeson, MIT Tedrake etc.

We agree that the introduction and the minimal citations to previous work were major weaknesses. The introduction section is substantially rewritten and extended. The number of referenced papers is tripled and include many more citations of RL and DL algorithms including by Panne, Atkeson and Tedrake.

2.     What is novel in this work? It seems that DDPG might not have been used for locomotion. But it is not clear why DDPG is particularly better than other RL, DL algorithms.

Continuous state space and action space limits the use of other RL/DL algorithms. The DDPG uses an actor-critic structure that can handle such problems and it is more efficient than using a stochastic policy gradient. But, the actor-critic structure requires training two networks that interact with each other at the same time, and so limiting the size and complexity of the DDPG network is important. This work provides a method for implementing DDPG into a locomotion control system by simplifying the state variables and using auxiliary controllers

3.     Why do all graphs look noisy? I was expecting to see periodic gaits that have periodic profiles

Disturbances are added at about 2 sec in figure 9, but periodic profiles still can be seen.

The scale for the y axis in fig 10 is small, so the difference between two periods looks large.

The sample frequency in figure 11 is per foot-step, so the curve is not periodic.

4.     An animation video showing the results would help substantially.

We have provided a screen recorded video.

Reviewer 2

1.       The objective of the paper is not defined well. It is argued that the main aim is to develop a controller for the exoskeleton. The developed controller has no association with the human walking control. The authors developed a nice controller that is able to stabilize the walking of a robot but there is no evidence that this controller can be used on an exoskeleton. If the controller has to be applied to Individuals with spinal cord injury (SCI), several validation experiments have to be delivered to compare the behavior of the controller and human. The main question is that if the controller can adapt itself to human body dynamics. If the controller is developed for more broad robotic applications, there should be a validation experiment regarding the comparison between this method and other walking controllers. The paper is descriptive of the proposed methodology where it fails to motivate why the robotic community should adopt yet another controller for biped robots.

The introduction and discussion sections are substantially rewritten and extended to address this point. The final goal for this controller is for a human/exoskeleton system. However, at this level of complexity, there is little difference between biped robot control and human/exo walking control if they have the same mechanical designs. This paper shows an implementation of deep reinforcement learning for biped locomotion control. Now that it is shown that DRL can handle such problems, a dedicated network can be trained for a specific system, human/exo or robot. The dynamics of the two systems will differ from each other (e.g. the control of human walking may need to transform the motor torques into muscle activations) but generally DNN treats it as a black box. New results are added to show that the system trained for a human of a particular size and mass also works when additional mass is added to the legs, representing the mass of the lower-limb exoskeleton. However, in future work, when the exoskeleton is finalized, the system will be retrained with the human/exo inertia properties.

2.     The only property of interest of the methodology is introduced as geometry scalability of the biped model and the controller is invariant to the changes in the body size.
Where this problem can be easily addressed by building a tuning table of the controller parameters for different morphologies.

The fact that the “controller is invariant to changes in the body size (and mass)” shows the stability margin of the controller. A tuning table would be sparse for this control system. In practice, we plan to use this controller as a starting point and retrain it for models of different human subjects (wearing the exo) to increase the stability of the system even further for a particular person. It is good to know that the results in this paper show that the system remains stable despite large changes in static loads and even for large impact forces, for example, when the person dons a backpack, picks up bag of groceries or is inadvertently pushed by another person. This is very important for a person/exo system in the real world.

3.     Depending on the objective of the paper the introduction section fails to provide a broad overview of the literature in this domain. Please extend the introduction part

The introduction section is substantially rewritten and extended. The number of referenced papers has been tripled.

4.     The main content of the paper is a description of the controller architecture which necessitates an addition of the block diagram for the proposed method.

Block diagrams have been added as figure 2 and figure 3, the Controller Architecture is explained in section 2.4. We rearranged the paper so the order of the figures has changed.

5.     Please improve figure 2

Figure 2 (now figure 4) is redrawn and the text describing it has been clarified.

6.     The main concern about the content of the paper is that it is written as a stand-alone paper where few citations have provided supporting the design decisions of the controller.

Again, we agree that this was a major weakness of the paper. The introduction was rewritten and the number of citations was tripled to set this work in its proper place within the literature.

Round  2

Reviewer 1 Report

The manuscript seems to be substantially improved. I think it is satisfactory and in my opinion worth publishing in Biomimetics.

Reviewer 2 Report

The introduction is fairly improved. The authors also added extra information all over the paper to make it more readable. I still think that illustrations can be improved further.

The major problem remaining in the paper is the language errors which is inserted to the paper in the review phase. It looks like that the authors did not proofread the paper since the name of the variables are misspelled and even some comments are forgotten to be removed.